The inheritance of female colour polymorphism in Ischnura genei (Zygoptera: Coenagrionidae), with observations on melanism under laboratory conditions

Sanmartín-Villar Iago sv.iago@uvigo.es
Cordero-Rivera Adolfo
ECOEVO Lab, Universidade de Vigo , Pontevedra , Galiza , Spain
Kelly Clint
Electronic publication date: 2016 Sep 1
Publication date: 2016
Volume: 4
Electronic Location ID: e2380
Received 2016 May 17; Accepted 2016 Jul 29
Copyright: ©2016 Sanmartín-Villar and Cordero-Rivera
Copyright year: 2016
Copyright holder: Sanmartín-Villar and Cordero-Rivera
License: This is an open access article distributed under the terms of the Creative Commons Attribution License, which permits unrestricted use, distribution, reproduction and adaptation in any medium and for any purpose provided that it is properly attributed. For attribution, the original author(s), title, publication source (PeerJ) and either DOI or URL of the article must be cited.
License URL: https://creativecommons.org/licenses/by/4.0/

Keywords: Phenotype, Fecundity, Colour changes, Reared generations, Laboratory effects, Odonata

Funding: Spanish Ministry with Competences in Science CGL2011-22629 CGL2014-53140-P BES-2012-052005 This work was funded by the Spanish Ministry with competences in science (grants CGL2011-22629 and CGL2014-53140-P, including FEDER funds and FPI grant BES-2012-052005). The funders had no role in study design, data collection and analysis, decision to publish, or preparation of the manuscript.

==============================
Current research on female colour polymorphism in Ischnura damselflies suggests that a balanced fitness trade-off between morphotypes contributes to the maintenance of polymorphism inside populations. The genetic inheritance system constitutes a key factor to understand morph fluctuation and fitness. Ischnura genei, an endemic species of some Mediterranean islands, has three female colour morphs, including one androchrome (male-coloured) and two gynochromes. In this study, we reared two generations of I. genei under laboratory conditions and tested male behavioural responses to female colour morphs in the field. We recorded ontogenetic colour changes and studied morph frequency in three populations from Sardinia (Italy). Morph frequencies of laboratory crosses can be explained by a model based on an autosomal locus with three alleles and sex-restricted expression, except for one crossing of 42 families with unexpected offspring. The allelic dominance relationship was androchrome > infuscans > aurantiaca. Old individuals reared in the laboratory exhibited different levels of melanism in variable extent depending on sex and morph. Results of model presentations indicate a male preference for gynochrome females and the lack of recognition of androchromes as potential mates. Aurantiaca females were the most frequent morph in the field (63–87%). Further studies in other populations and islands are needed to understand the maintenance of this polymorphism.

Introduction

Phenotypic variability is biologically important because it increases adaptive opportunities, and provides a potential avenue for speciation. This kind of variability could be promoted by plasticity (e.g., Radwan (1993)) or be inherited (e.g., polymorphism). Polymorphism occurs when alternative intra-population morphs are found at higher frequencies than expected by mutations alone (Ford, 1957). Colour polymorphism is a common phenomenon in the Odonata, particularly in coenagrionids where females exhibit different colour morphs (Cordero & Andrés, 1996; Fincke et al., 2005). The maintenance of this kind of phenotypic variability suggests a balance between the costs and benefits of each phenotype based on different mechanisms of frequency- and/or density-dependent selection (Roulin & Bize, 2007). The adaptive significance of female body colour variation is currently interpreted as an evolutionary response to male reproductive harassment. In this scenario, each female colour morph is balanced by different costs and benefits across fluctuations in the intensity of male harassment, which is frequency-dependent (Van Gossum, Sherratt & Cordero-Rivera, 2008). In all polymorphic Coenagrionid species so far studied, one of the female morphs presents similar body colouration as the conspecific male (androchrome) and other/s show a different colour (gynochromes). This variation is controlled by one autosomal locus with 2–3 alleles and sex-restricted expression, where only females show colour morphs (Johnson, 1964; Johnson, 1966; Cordero, 1990; Andrés & Cordero, 1999; Sánchez-Guillén, Van Gossum & Cordero-Rivera, 2005). However, the dominance of these alleles does not follow a single mechanism among species (including those of the same genus). The androchrome allele is recessive in Ischnura damula (Johnson, 1964), I. demorsa (Johnson, 1966), and Ceriagrion tenellum (Andrés & Cordero, 1999); has been suggested to be recessive in Ischnura senegalensis (Takahashi et al., 2014); but is dominant in Ischnura graellsii (Cordero-Rivera, 1990) and I. elegans (Sánchez-Guillén, Van Gossum & Cordero-Rivera, 2005).

The island bluetail damselfly, Ischnura genei (Rambur, 1842), is a species restricted to the Mediterranean islands (Tuscan archipelago and the Tyrrhenian and Maltese islands) (Boudout & Kalkman, 2015). This species cohabits with I. elegans in Giglio Island (Boudout & Kalkman, 2015) where both might hybridize (see Sánchez-Guillén et al., 2013c). For some time, I. genei was considered a subspecies of I. elegans, but currently it is considered as a valid species (Dumont, 2013). Nevertheless, very little is known about its biology, and basic information remains undescribed, like its ontogenetic colour change and the inheritance of its colour morphotypes.

In this paper, we test the prediction that the maturation and inheritance of the colour polymorphism of I. genei should be similar to its closest relatives (I. graellsii and I. elegans) and that the dominance of the alleles that determine its polymorphism follows the rule androchrome > infuscans > aurantiaca. Infuscans and aurantiaca represent the gynochrome morphs (olive to brown and orange-brown without humeral stripe when immature, respectively). We named the morphs using the terminology of I. graellsii to avoid confusing terms as in I. elegans, in which the same morph was named differently according to its age (rufescens, infuscans-obsoleta, or rufescens-obsoleta; e.g., Sánchez-Guillén et al., 2011; Van Gossum et al., 2011).

Table 1 Results of the egg clutches obtained from field-collected females from Riu Mannu (Tula, Sassari, Sardinia–Italy).

Hatching time: number of days between oviposition and first hatch; Fecundity: total number of eggs laid by each female. Values are mean ± SE.

P females	Hatching time (days)	Fecundity	Unfertile eggs	Fertility(%)	
Androchromes	
A	14	78	22	71.79	
B	12	174	69	60.34	
C	12	217	15	93.09	
D	14	188	38	79.79	
E	12	135	15	88.89	
F	12	138	23	83.33	
Total	12.7 ± 0.42	155.0 ± 19.92	30.3 ± 8.46	79.5 ± 4.87	
infuscans	
G	14	245	17	93.06	
H	14	145	44	69.66	
I	14	231	20	91.34	
J	14	373	15	95.98	
K	14	83	49	40.96	
Total	14.0 ± 0.00	215.4 ± 49.21	29.0 ± 7.23	78.2 ± 10.41	
aurantiaca	
L	17	185	74	60.00	
M	14	228	6	97.37	
N	12	337	3	99.11	
O	12	114	2	98.25	
P	9	175	1	99.43	
Q	12	97	6	93.81	
R	9	272	0	100.00	
S	9	259	47	81.85	
T	14	147	3	97.96	
Total	12.0 ± 0.91	201.6 ± 26.31	15.8 ± 8.77	92.0 ± 4.41	
Total average	12.7 ± 0.45	191.1 ± 17.92	23.5 ± 5.05	84.8 ± 3.68	
TOTAL		3,821	469		

Material and Methods

Rearing process

Six androchromes, five infuscans and nine aurantiaca females (Parental generation (P), N = 20) were captured while in copula on 19 and 24 of August of 2011 from Riu Mannu (40.687294 N, 8.989478 E), near Tula (Sardinia—Italy). After mating, females were placed in cups with wet filter paper to obtain their eggs. Clutches were thereafter maintained in water and checked daily to record hatching date. Larvae were reared in the laboratory in accordance with the previously described methodology (Gossum, Sánchez-Guillén & Rivera, 2003; Piersanti et al., 2015).

Two generations of I. genei were raised in the laboratory. Data from females with the identification codes K, S and T (see Table 1) were excluded because they produced a very low number of offspring. Two additional females (F and R) also produced too few offspring and proportions could not be tested, but their progeny were used as parents in the next generations. One month after hatching, 947 F1 larvae (field-collected females’ offspring) were kept individually in small plastic cups and then transferred to cells of ice cube trays with a plastic net as the bottom. This structure allowed for the change of water in groups, the supply of continuous oxygen via an air pump (one for every three ice cube trays), and the elimination of debris by gravity. To prevent the mix of larvae, cells were not completely filled. Spring water was used. Larvae were fed daily ad libitum with Artemia nauplii.

Final instar larvae were transferred to individual 1 L plastic containers filled with 250 mL of water. A wooden cooking stick (length = 20 cm, diameter = 0.33 cm) was placed inside containers as a substrate for metamorphosis. A net covering the container prevented the escape of adults and supported the wooden stick. After metamorphosis, all water was removed to avoid drowning the newly emerged adults. One day after metamorphosis, imagoes were moved into wooden insectaries (50 × 50 × 50 cm). A glass covered the top and the upper third frontal side of the insectaria. Interior wooden surfaces of the insectaries were covered with aluminium foil, which prevents escape because damselflies perceive the foil’s reflectance as the brightness of an open area and thus avoid it (Johnson, 1965). Wooden sticks, fine branches, and nets glued to the top of the insectary provided perches to enrich the environment and limit agonistic interactions. The insectaries were illuminated by sunlight from laboratory windows and one 60W incandescent bulb positioned 10 cm above each insectary. Adult damselfiles were fed adult Drosophila melanogaster flies. A constant supply of flies was maintained by having a bottle with food and egg-laying substrate for the flies in each insectary. A container of water covered with a net was introduced to increase humidity inside the insectary. A maximum of 10 imagoes lived together in each insectary. Individuals in the insectaries were separated to prevent reproductive or agonistic interactions, taking into account sex and age (according to their maturation colour).

The second generation was obtained from 28 laboratory matings and reared under the same conditions. Four new wooden insectaries, similar to those previously described but with transparent plastic panels instead of glass, were also employed.

Parental genotypes of P and F1were inferred based on the phenotypes of their offspring.

Herein we follow the notation of Cordero (1990) for the polymorphism locus (p) and each allele: androchrome (pa), infuscans (pi) and aurantiaca (po).

Fecundity and fertility of field-collected females

One month after the last larval hatch, eggs laid by field females were counted under a binocular microscope. Fecundity was measured as the total number of eggs laid. Unhatched eggs without an embryo were considered sterile, whereas those that had an embryo inside but did not hatch were considered fertile, because the paper used as oviposition substrate might have impeded some larvae from hatching. These numbers were used to calculate fertility.

Colour maturation

Photographs were periodically taken to have a record of the thorax and eighth abdominal segment (S8) colourations.

Female morph frequencies in the field

The proportion of female morphs was estimated by counting adults from three populations in 2008, 2011, and 2012 on the island of Sardinia (Italy). Adults were captured, and their sex, age, and morph were recorded. All were marked to avoid counting them twice, and then released at the place of capture.

Morph choice by males

The androchrome females of this species were identical to males in body colouration and ontogenetic colour changes (see ‘Results’). In August 2012, we studied the population from Riu Mannu to test whether males are able to distinguish androchrome females from males and whether males have a preference for gynochromes.The experiment followed the protocol of previous studies (Cordero, Santolamazza-Carbone & Utzeri, 1998), but used only live models to take into account behavioural differences. Models were mature males or females tethered to a wooden stick using a fine wire, which allowed them to fly. Models were presented to males by approaching them and leaving the model to perch near the focal male. The behaviour of test males was scored as no response, approach to the model, attempt to grasp the model in tandem, and tandem. The order of model presentation was chosen randomly in the first trial and repeated in subsequent trials. Each model was used until 10 males responded. Males that did not respond (5–7 for each individual model) were excluded from the analyses. Whenever possible, each male was marked to avoid testing it twice. Three models were used for each phenotype.

Statistical methods

Hatching time for eggs from the P generation were analysed using a Kruskal Wallis test. We used ANOVA to determine whether androchrome and gynochrome females differed in fecundity. Fertility proportions were analyzed using a GLM with binomial errors, corrected for overdispersion. Morph was the only factor entered in both analyses. In the inheritance experiment, the observed frequencies of phenotypes were compared with the expected frequencies using a Chi square test. Some infuscans and androchromes died before sexual maturation and could not therefore be assigned to a morph (all aurantiaca females can be easily identified since emergence). In the second generation, we estimated how many of these females might be infuscans or androchromes, based on the proportion of these morphs in the remaining females of each family. No statistical analyses were performed on progenies where only one morph was expected and found. Sex-ratio differences between morph offspring were analysed by GLM with binomial errors and Tukey test. Male behaviour towards models was analysed by a logistic regression, with model phenotype and model identity (each specimen used in the experiment) entered as predictor variables. Only six males were able to get the model female in tandem (out of 120 tested). Therefore, we used a binomial response variable: approach versus tandem (including tandem attempts and successful tandems). Data were analysed with the statistical package R version 3.2.3 (R Core Team, 2016), Genstat 18th edition (VSN International, 2015) and with xlStat 2016 (www.xlstat.com).

Results

Fecundity and fertility of field-collected females

There were no differences among female morphs in egg hatching time (H = 4.61, p = 0.100), in the number of eggs laid (F = 1.03, p = 0.322), or in the proportion of fertile eggs (GLM with binomial errors, deviance ratio = 1.20, p = 0.324, Table 1).

Figure 1 Mature wild individuals from the field.

Mature wild individuals of I. genei from Sardinan populations. No melanism was observed in the field. (A) male; (B) androchrome; (C) infuscans; (D) aurantiaca. Photos: Adolfo Cordero-Rivera.

Colour morphs and maturation

Mature field-collected females showed three colour phenotypes (Fig. 1), equivalent to the morphs described for I. elegans (Sánchez-Guillén, Van Gossum & Cordero-Rivera, 2005) and I. graellsii (Cordero, 1990).

One day after emergence males presented a range of green colours, tending toward green-yellow, in the thorax. The thoracic green colour became darker with the maturation process and turned to blue in older individuals (Fig. 2).

Figure 2 The ontogeny of colour changes in the female morphs of I. genei.

Age refers to the mean values for colour changes under laboratory conditions.

Different thorax colours were observed in androchrome females one day after their emergence (pink (more frequent), sky blue, or green-yellow; Fig. 2). Pink individuals became sky blue between one and three days after emergence. Greenish females were found in the progeny of female 412 (six out of 18 females) and 499 (one out of three). Three days after emergence, androchromes showed a similar body colour to males (Fig. 2).

The thorax of immature infuscans females was violet and the light-coloured antehumeral stripe was usually wider than in males and androchromes (Fig. 2). The thorax of mature infuscans females was olive-green but then turned brownish at old age (more than ten days after their emergence). Most infuscans females showed a variable black spot on the eighth abdominal segment.

Immature aurantiaca females presented pale orange and pink thorax after emergence, with one medio-dorsal black stripe, and no black humeral stripes. Orange individuals became pink two days after emergence. Thoracic pink colour turned greenish or brown around five days of age (Fig. 2), and dark brown humeral stripes developed. These two alternative colours (green or brown), observed both in the laboratory and in the field, remained until death, showing different processes of darkening and no transitions between them. This suggests that there are two mature colours in the aurantica morph. In one case, the brown colour of the brownish individuals covered the thoracic dorsolateral region, with only the metepimeron remaining light in colour. In the other case, greenish thoraxes presented one brown spot in the anterior part of the mesothorax that suffered a darkening process with age at the same time as the humeral stripe (Fig. 2).

In the laboratory, adult damselflies presented a melanic process across maturation (Fig. S1). The first signal was the appearance of a brown spot in the anterior part of the mesothorax in early mature individuals (column Mature 1 and Mature 2 for aurantiaca females in Fig. S1). This spot was only seen in the field in the aurantiaca morph (Figs. 1 and 2). Melanism continued covering the dorso-lateral thorax part of males and androchromes, forming a diagonal stripe in infuscans but it did not affect aurantiaca females. In reared males and androchromes, S8 showed melanic spots (Fig. S2). These black spots were present in immature individuals and became wider during their maturation but never completely replaced the blue colour. Gynochromes presented a wide range of colour variability in S8 but in all cases the blue colour disappeared with maturation. Black spots of S8 were wider in aurantiaca than in infuscans females.

Pruinescence was observed in old individuals (more than 20 days) when thoracic colours lost vivacity. A whitish dust appeared in dorsal parts (head, dorsal carina, coxae and in the three last abdominal segments).

Proportions in the first generation (F1)

A total of 603 adults successfully emerged in the first generation from 21 Dec 2011 until 30 May 2012. The segregation of phenotypes was in agreement with the allele dominance pa > pi > po. Proportions were as expected from the inheritance hypothesis, with the exception of a minority of individuals, which could be the offspring of females mated multiply in the field (Table 2).

Table 2 The segregation of female colour morphs in the F1 generation.

♀P: females from the field; Parental genotypes: alleles of each progenitor (female and male); ♀F1: number of F1 females; Observed: number of F1 identified females by morph and percentage in respect to the total number of females of the same cross (between brackets); Expected: percentages expected under the allelic order of dominance pa > pi > po. Observed and expected frequencies were compared with a χ2 test, and the associated p-value is presented.

♀P	P genotypes	Sex ratio (♂:♀)	♀F1	Observed (N and %)	Expected (%)	χ2	p	
	♀	♂			A	I	O	A	I	O			
Androchrome	
B	papo	papa	0.14	7	6 (85.7)	0 (0)	1a (14.3)	100	0	0			
C	papo	papo	0.48	31	25 (80.6)	0 (0)	6 (19.4)	75	0	25	0.00	1.00	
D	papo	popo	0.84	19	9 (47.4)	0 (0)	10 (52.6)	50	0	50	0.05	0.819	
E	papo	pipo	1.36	14	8 (57.1)	3 (21.4)	3 (21.4)	50	25	25	0.19	0.665	
infuscans	
G	pi∕opi	pi∕opi	1.42	31	1a (3.2)	29 (93.5)	0 (0)	0	100	0			
H	pipo	pi∕opo	0.38	8	0 (0)	2 (25.0)	6 (75.0)	0	50	50	2.00	0.157	
I	pipo	pipo	0.67	9	0 (0)	6 (66.7)	3 (33.3)	0	75	25	0.00	1.00	
J	pipo	popo	2.17	6	1a (16.7)	2 (33.3)	3 (50)	0	50	50	0.00	1.00	
aurantiaca	
L	popo	pipo	1.5	6	0 (0)	3 (50.0)	3 (50.0)	0	50	50	0.00	1.00	
M	popo	popo	1.31	14	0 (0)	1a (7.14)	13 (92.9)	0	0	100			
N	popo	popo	0.79	34	0 (0)	0 (0)	34 (100)	0	0	100			
O	popo	popo	0.86	14	0 (0)	2a (14.3)	12 (85.7)	0	0	100			
P	popo	popo	1.67	15	1a (6.7)	0 (0)	12 (80.0)	0	0	100			
Q	popo	popo	1.06	18	0 (0)	0 (0)	18 (100)	0	0	100			
Notes.

a Possible progeny of a second mate in the field (these values were not included in the calculation of proportions).

When offspring were composed of two colour morphs, they presented a ratio of 1:1 or 3:1 (A:I or I:O). The progeny of female E produced all three phenotypes in a proportion 2:1:1 (A:I:O), which is the expectation when two heterozygotes for different alleles mate. The allele po was present in at least in 27 out of 34 parental individuals (it was not possible to identify all alleles in eight individuals).

Overall, F1 sex-ratio (male:female) was not biased (1.01:1) (Table 2), nor were differences among morphs (A = 0.74:1; I = 1.22:1; O = 1.09:1; deviance ratio = 1.32, p = 0.304).

Proportions in the second generation (F2)

A total of 1,105 imagoes emerged from 17 April to 28 December 2012. All F2 proportions (N = 28 families; Table 3) followed the expected allelic dominance except in one family (cross between female 243 and male 66), that produced four unexpected androchrome offspring. All but two F1 crossed individuals (N = 56) showed one po allele (in these two cases it was not possible to identify all alleles).

Table 3 The segregation of female morphs in the F2 generationn.

F1 individuals’ codes include an ordinal and a letter identifying the female progenitor; N females: total number of females whose morph was scored.

Parental genotypes	F1 individuals	Sex ratio (♂:♀)	♀F2	Observed (N and %)	Expected (%)	χ2	p	
♀	♂	♀	♂			A	I	O	A/I	A	I	O			
Androchromes	
papa?	papa?	499 C	491 P	0.67	12	3 (25.0)	0 (0)	0 (0)	9 (75.0)	100	0	0			
papi	papo	412 G	E4.1 A	0.00	33	18 (54.54)	9 (27.27)	0 (0)	6 (18.18)	75	25	0	0.00	1.00	
papo	pipo	190 E	208 G	0.40	20	6 (30.00)	1 (5.00)	5 (25.00)	6 (30.00)	50	25	25	3.00	0.223	
		200 C	193 G	0.48	25	2 (8.00)	6 (24.00)	9 (36.00)	7 (28.00)	50	25	25	3.00	0.223	
		510 F	514 G	0.00	11	3 (27.27)	2 (18.18)	4 (36.36)	2 (18.18)	50	25	25	3.00	0.223	
papo	popo	161 C	180 P	1.32	19	5 (26.32)	0 (0)	7 (36.84)	7 (36.84)	50	0	50	1.31	0.251	
		258 F	242 P	0.80	10	8 (80.00)	0 (0)	1 (10.00)	1 (10.00)	75	0	25	0.00	1.00	
		272 C	192 P	0.67	15	7 (46.67)	0 (0)	4 (26.67)	4 (26.67)	75	0	25	0.00	1.00	
		368 C	361 J	0.43	7	3 (42.86)	0 (0)	1 (14.29)	3 (42.86)	50	0	50	3.57	0.059	
infuscans	
pipo	pipo	293 J	301 G	0.78	9	0 (0)	7 (77.78)	2 (22.22)		0	75	25	0.00	1.00	
pipo	popo	222 L	242 P	1.35	17	0 (0)	6 (35.29)	10 (58.82)		0	50	50	1.00	0.317	
		257 L	180 P	0.82	22	0 (0)	6 (27.27)	14 (63.63)	2 (9.09)	0	50	50	1.64	0.201	
		285 G	338 O	0.42	23	0 (0)	11 (47.83)	11 (47.83)		0	50	50	0.00	1.00	
		286 G	281 O	0.20	20	0 (0)	8 (40.00)	11 (55.00)		0	50	50	0.47	0.491	
		325 O	378 Q	0.73	11	0 (0)	6 (54.54)	5 (45.45)		0	50	50	0.09	0.763	
		476 I	473 J	0.00	13	0 (0)	6 (46.15)	7 (53.85)		0	50	50	0.08	0.782	
aurantiaca	
popo	papo	E9.3 H/O	E4.1 A	0.17	23	7 (30.43)	0 (0)	14 (60.87)	2 (8.69)	50	0	50	1.09	0.297	
popo	pipo	243 H	66 E	0.40	15	4a (26.67)	1 (6.67)	8 (53.33)	2 (13.33)	0	50	50			
		255 O	208 G	0.45	22	0 (0)	6 (27.27)	13 (59.09)	1 (4.54)	0	50	50	1.8	0.180	
		268 M	66 E	1.00	12	0 (0)	4 (33.33)	5 (41.67)	1 (8.33)	0	50	50	0.00	1.00	
		291 R	201 O	0.13	15	0 (0)	4 (26.67)	11 (73.33)		0	50	50	3.27	0.071	
		508 R	514 G	0.00	17	0 (0)	7 (41.18)	10 (58.82)		0	50	50	0.53	0.467	
		333 R	342 I	0.00	10	0 (0)	3 (30.00)	7 (70.00)		0	50	50	1.6	0.206	
popo	popo	9 J	8 O	0.75	12	0 (0)	0 (0)	12 (100)		0	0	100			
		168 P	192 P	0.75	16	0 (0)	0 (0)	13 (100)		0	0	100			
		262 L	180 P	0.89	19	0 (0)	0 (0)	18 (100)		0	0	100			
		274 O	242 P	0.38	13	0 (0)	0 (0)	12 (100)		0	0	100			
		471 R	473 J	0.00	10	0 (0)	0 (0)	10 (100)		0	0	100			
Notes.

a Case where the phenotype is not expected.

A/I Androchrome or infuscans females that died before maturation and could not be assigned to a morph

In contrast with F1, the sex-ratio was deviated towards females in F2 generation (0.53:1). This female-biased sex-ratio occurred irrespective of maternal morph (A = 0.53:1; I = 0.61:1; O = 0.41:1; deviance ratio = 0.36, p = 0.704).

Table 4 The frequencies of female colour morphs in three Sardinian populations of I. genei.

The total number of mature adult males and females is given, together with the proportion of each female morph. There are no reliable cues to distinguish androchrome and infuscans females when they are immature because both can be violet. S8B indicates females with S8 mostly blue (as in Fig. S2I), which usually mature as androchromes, whereas S8BB refers to females with blue and black in S8 (as in Fig. S2J), which always mature as infuscans.

	Population	
	Riu Foxi, Campus (Villasimius, Cagliari)	Riu Mannu, Tula (Sassari)	Riu Mannu, Tula (Sassari)	Riu de Li Saldi, Lu Lamoni (Olvia Tempio)	Riu de Li Saldi, Lu Lamoni (Olvia Tempio)	
Latitude (°N)	39.137309	40.687294	40.687294	41.127444	41.12744	
Longitude (°E)	9.489396	8.989478	8.989478	9.087461	9.087461	
Altitude (m)	7	163	163	4	4	
Date	23 Aug 2008	19, 24 Aug 2011	21, 23 Aug 2012	23, 26, 28 Aug 2011	20, 25 Aug 2012	
Sex-ratio (M/F)	1.14	1.60	1.30	2.02	1.10	
Mature adults						
Males	64	235	181	106	32	
Females	56	79	86	47	28	
Androchromes	0.232	0.165	0.163	0.021	0.107	
infuscans	0.143	0.165	0.081	0.106	0.036	
aurantiaca	0.625	0.671	0.756	0.872	0.857	
Young adults						
Males	–	22	28	1	0	
Females	–	82	75	6	1	
violet-S8B	–	0.024	0.080	0.000	0.000	
violet-S8BB	–	0.098	0.160	0.500	0.000	
aurantiaca	–	0.878	0.760	0.500	1.000	

Female morph frequencies in the field

Table 4 shows sex-ratio and the frequency of female morphs in three populations from Sardinia. In all cases the aurantiaca morph was the most common (63–87% among mature females), androchromes were 2–23%, and infuscans 4–17%. Two of the populations were sampled in two consecutive years, showing little changes in frequencies.

Morph choice by males

The results of the experiment of model presentations to mature males indicated that males do not discriminate between androchromes and other males, and showed preference for gynochromes (Fig. 3). A logistic regression indicates that model phenotype affected male behaviour (χ32=11.65, p = 0.009) but model identity (each one of the three individuals used per phenotype) does not (χ82=6.05, p = 0.642). Males showed low sexual interest in other males and androchromes, without differences between these phenotypes (χ12=0.38, p = 0.538), and higher sexual interest to both gynochromes, but without differences between them (χ12=0.22, p = 0.640).

Figure 3 The response of I. genei males to live models.

Males that simply approached the model were scored as not showing sexual interest. Males that tried or achieved tandem with the model were scored as showing a sexual response. The response to male and androchrome models was clearly different to both gynochrome morphs (infuscans and aurantiaca).

Discussion

Our results indicate that the colour polymorphism of I. genei is homologous to that of I. elegans (Sánchez-Guillén, Van Gossum & Cordero-Rivera, 2005) and I. graellsii (Cordero, 1990), with one androchrome phenotype and two gynochrome phenotypes: infuscans and aurantiaca. The allelic dominance is the same as in the cited species: androchrome > infuscans > aurantiaca.

Colour morphs of I. genei and laboratory melanism

In our experiments, most androchrome I. genei females had a violet thorax before sexual maturation, which was indistinguishable from the colouration of the thorax of immature infuscans females; a colour-change pattern that is also seen in I. elegans (Sánchez-Guillén, Van Gossum & Cordero-Rivera, 2005; Cordero-Rivera, 2015). Nevertheless, six females were pale green and did not develop violet colouration, following the same ontogenetic colour changes as males. This observation indicates that there are two phenotypes which mature into androchromes, as previously described for I. elegans (Sánchez-Guillén, Van Gossum & Cordero-Rivera, 2005). These females were obtained in crosses where both parents had at least one androchrome allele and might be homozygous for that allele, as previously suggested for I. elegans (Sánchez-Guillén, Van Gossum & Cordero-Rivera, 2005). In the laboratory, both males and androchromes presented a green thorax as the final bright colouration in their ontogeny (before becoming dark). In contrast, individuals from the field showed blue colouration when mature, as is typical of I. elegans (Cordero-Rivera, 2015).

Mature infuscans females were olive-green in the laboratory, and some achieved a brownish thorax. The other gynochrome phenotype (aurantiaca) was pinkish shortly after emergence, and clearly identifiable. Nevertheless, both in the laboratory and in the field we observed that mature aurantiaca females exhibit two colour variants. Some females became greenish, with secondarily developed brown humeral lines, recalling the gynochrome females of some other Ischnura species like I. rufostigma (Sanmartín-Villar, Zhang & Cordero-Rivera, 2016). Other aurantiaca females became brownish, like the mature colouration of the aurantiaca of I. graellsii (Cordero-Rivera, 1987) and I. elegans (Sánchez-Guillén, Van Gossum & Cordero-Rivera, 2005). We have observed both aurantiaca variants in populations of I. graellsii in NW Spain. We hypothesize that the expression of a second gene at maturity in aurantiaca females can modify a pigment pathway (see Chauhan et al., 2014) and determine these two phenotypes. This possibility needs further study.

Our laboratory breeding produced melanic individuals (Fig. S1), which complicated morph assignment. Melanism was previously found in other captive-reared damselflies in our laboratory, notably in Coenagrion scitulum (Cordero, Santolamazza-Carbone & Utzeri, 1995) but also in other species (Enallagma cyathigerum, Pyrrhosoma nymphula, Platycnemis latipes and P. acutipennis; A Cordero-Rivera, pers. obs., 2000) although not in Ischnura (I. elegans, I. graellsii, I. hastata , I. pumilio, I. rufostigma, I. saharensis; Cordero-Rivera, 1990; Sánchez-Guillén, Van Gossum & Cordero-Rivera, 2005; Lorenzo-Carballa & Cordero-Rivera, 2009; RA Sánchez-Guillén, pers. comm., 2009; RA Sánchez-Guillén, pers. obs., 2009). Similar melanic colouration in several Enallagma species was assumed as an effect of the incomplete UV radiation that larvae and tenerals received under captive conditions (Barnard et al., 2015). In our case, the only difference between I. genei and the other species was the food supply. We fed I. genei only with Artemia nauplii during the whole larval period, while in the other cases last instar larvae received a supplement of Tubifex worms or adult Artemia (which has different nutrient composition). However, Barnard et al. (2015) suggest that the nutritional supply cannot produce melanism. In our case, melanism only appeared in old individuals while in the other cited cases the first adult colouration was affected by melanism.

The darkening of S8 in males and androchromes (Fig. S2) was not observed in the field even in old individuals. In the laboratory, eighth abdominal segment dark spots were already present in immature individuals. The loss of blue colour in the final abdominal parts constitute the normal maturation process of gynochromes of this species. In I. rufostigma some males and androchromes present a large black spot in segment eight irrespective of age (Sanmartín-Villar, Zhang & Cordero-Rivera, 2016). This kind of colour modification is due to a distributional change that the spheres of the endoplasmic reticulum of the pigment cells suffer, which also constitute the mechanism of colour change in species that show temperature-related colour changes (Veron, O’Farrell & Dixon, 1974; Prum, Cole & Torres, 2004).

The inheritance mechanism

The observed proportions in our breeding followed the expected allelic dominance previously found in I. graellsii (Cordero, 1990) and I. elegans (Sánchez-Guillén, Van Gossum & Cordero-Rivera, 2005). Nevertheless, five out of 15 crosses in the F1 generation and one out of 28 crosses in the F2 generation did not follow the expected allelic hierarchy. The unexpected female morphs from F1 crosses (one or two females per cross) can easily be explained by multiple paternity. The parental generation females were collected from the field and likely mated several times before being collected. In fact, paternity of the last male is usually not 100% in coenagrionids (79% of the offspring was sired by the last male in Ischnura elegans (Cooper, Miller & Holl, 1996); 92–100% in Ischnura graellsii Cordero & Miller, 1992; and 95% in Enallagma hageni Fincke, 1984).

The unexpected presence of four androchrome females in the offspring of aurantiaca female 243 (popo) and male 66 (pipo) (Table 3) could be explained by: (i) a mislabelled container during larval breeding, (ii) a mistake in morph identification, or (iii) an unknown genetic mechanism. We are confident that all larvae were correctly assigned to their family because individuals were followed from their isolation in the larval stage until their death, including the mating process, and therefore no unnoticed matings could have occurred. We tried to check the possibility of mislabelled larvae using a paternity test with microsatellites but the ambiguity of genetic results did not allow us to confirm the family of origin of these problematic specimens. However, the genetic results were compatible with the presumed family of origin. We also think that morph identification was correct, even if the melanism somewhat obscured some cues. Morph identification was done in a conservative way, so that females were only assigned to a morph when no doubts existed (in fact some could not be identified; see A/I column in Table 2). Androchrome and infuscans females were mainly classified by the extent and persistence of S8 blue colouration. Melanism affected S8 colouration. However, while all infuscans females lost their blue colouration at around 6 days of age, androchromes presented black marks in the middle of the blue spot at around 15 days of age and never lost all the blue colour. Photographs taken along the individual maturation suggest that these four females were correctly identified as androchromes. Exceptional females with intermediate colours between androchrome and infuscans morphs were observed in the field (one individual in I. graellsii (Cordero, 1992) and another in I. elegans; A Cordero-Rivera, pers. obs., 2016). It is therefore possible that the unexpected “androchrome” females in the progeny of female 243 are the result of an unknown genetic mechanism, producing an intermediate phenotype, and do not invalidate our inheritance hypothesis.

We found that the sex-ratio of our second generation was female biased although we employed the same breeding methodology in both generations. As in Bots et al. (2010), our experiment exclusively employed Artemia as larval food. Bots et al. (2010) experienced problems with larval survivorship and adult maturation. The parental stress hypothesis proposes that females subjected to low quality food intake produce maladaptive maternal effects in offspring (Rossiter, 1991; Rossiter, 1996; Vijendravarma, Narasimha & Kawecki, 2010) affecting egg quality and/or embryo or larval survivorship. In our case, the preponderance of females in our second generation suggests that male larvae may have been more susceptible to maternal effects due to the differences between sexes in larval activity and/or time needed to emerge (Debecker et al., 2016), or were more susceptible to cannibalism during early larval development (before they were isolated).

The maintenance of colour morphs

The maintenance of female colour polymorphism in odonates, and particularly in Ischnura, has been intensively discussed in recent years (e.g., Takahashi et al., 2010; Iserbyt et al., 2013; Sánchez-Guillén et al., 2013b; Le Rouzic et al., 2015). Several hypotheses have been proposed based on frequency- and density-dependent selection. In one hand, some authors propose that male-mimicry by androchromes is at the centre of the maintenance mechanism because they avoid male harassment in high male densities but loss mating opportunities when males are scarce (Robertson, 1985; Hinnekint, 1987). In the other hand, other authors suggest that males learn to recognize as female the commonest morph in the population (Fincke, 2004). Recent studies indicate that both mimicry and learning are involved in the maintenance of female colour polymorphism in odonates (Sánchez-Guillén et al., 2013a). The data available for I. genei are not enough for a comprehensive discussion and therefore we will concentrate on whether male-mimicry is likely to have a main role or not in this species.

The experiment of models presentation clearly demonstrated that most males do not recognize androchromes as potential mates, showing a clear preference for gynochromes (52% of sexual response versus 11% for androchromes). Therefore, the pre-requisites for the male-mimicry hypothesis (Robertson, 1985) are held. It is important to consider that this experiment was done on a population where androchromes were only 16% (Table 4). If males learn to prefer the most common morph, then we expect a clear preference for aurantiaca females (76% in the population). Nevertheless, we found that males showed similar preference for the rarest infuscans (8%) and the most common aurantiaca. This suggests that males do not discriminate between both gynochrome females (see Xu et al., 2014). We hypothesize that mature aurantiaca and infuscans females cannot be distinguished by males. Further experiments with models in populations with contrasting frequencies would be illuminating.

We did not find differences in fecundity between female morphs as in previous studies on other species Bots et al., 2010; as Barnard et al., 2015; Galicia-Mendoza, 2015 between others). However, other studies (Svensson, Abbott & Härdling, 2005; Takahashi & Watanabe, 2010) showed differences in fecundity among morphs. We found a trend in fertility that suggests a higher value in aurantiaca females. If that trend were real, it could explain the preponderance of aurantiaca females in both field and laboratory conditions. Female coenagrionids maximize their reproductive success by minimizing inter-clutch interval rather than maximizing clutch size both in the field (Banks & Thompson, 1987) and the laboratory (Cordero, 1991). This suggests that fecundity/fertility differences between ovipositions performed in a single day might be biologically irrelevant if morphs differ mainly in lifetime number of clutches (Cordero, 1994) or if they follow different strategies to lay a different number of eggs through time (RA Sánchez-Guillén, pers. comm., 2015). Data on the frequencies of female morphs in other populations and an estimation of lifetime fecundity under field conditions are needed to model the maintenance of this polymorphism in I. genei.

In conclusion, our results suggest a common allelic dominance system in the European Ischnura species. In addition, we found that the male-mimicry hypothesis is the best explanation for the maintenance of polymorphism in I. genei. This opens the possibility that colour polymorphism is in fact an array of different phenomena, and suggest that a single—common- explanation is unlikely to be applicable to all species.

Supplemental Information

Data S1 P generation clutches; emergences in the laboratory; sex-ratio; models presentation

Click here for additional data file.

Figure S1 Ontogenic thorax colour change in reared individuals

Ontogenic thorax colour change for sex and morph in reared individuals. Thoraxes are positioned in a lateral point of view, with the head to the right and the abdomen on the left. Two thoraxes in the same stage of the same morph (e–f, p–r, q–s) show colour variation and not change. Immature: one day after emergence; Mature 1, early mature individual (5–7 days after emergence); Mature 2, older mature individual (more than 7 days after emergence); Old: individual with more than 15 days.

Click here for additional data file.

Figure S2 Dorsal view of final abdominal segments

Dorsal view of final abdominal segments showing the pattern variance in reared immature individuals. The last two images of each row correspond to the same individual at different times to appreciate the ontogenic change. Individuals were placed in this order to show the common development of the melanic pattern. Males: upper row. Androchromes: lower row. (A–B): complete blue coloured S8. (C) black colour started to develop as two dorsal arches in the S8 (anterior and posterior) with a peak in the middle towards the inner region of the segment. (D–G) appearance of two symmetrical dots around carinae and three dots in the lateral part of the S8; a longitudinal black stripe connected the superior and inferior black arches due to the elongation of the middle peaks; this stripe became wide in the basal region; the two dots and the basal regions became connected. Not all individuals showed the complete development. Individuals could present the S8 colouration in an advanced pattern one day after emergence and/or maintain one intermediate pattern until death.

Click here for additional data file.

Thanks to Dalia Ivette Galicia-Mendoza and María Olalla Lorenzo-Carballa for their advice and help with the laboratory rearing and for their ideas and participation in the construction of the new rearing structures. Thanks to Genaro da Silva-Méndez and Rosa Ana Sánchez-Guillén for their support during the genomic analyses. Thanks also for the comments and corrections performed by RASG, Clint Kelly, Jacek Radwan and an anonymous reviewer. Austin Kent Moore and Conor Higgins improved our English.

Additional Information and Declarations

Competing Interests

Author Contributions

Data Availability

The authors declare there are no competing interests.

Iago Sanmartín-Villar performed the experiments, analyzed the data, contributed reagents/materials/analysis tools, wrote the paper, prepared figures and/or tables.

Adolfo Cordero-Rivera conceived and designed the experiments, performed the experiments, analyzed the data, contributed reagents/materials/analysis tools, wrote the paper, prepared figures and/or tables, reviewed drafts of the paper.

The following information was supplied regarding data availability:

The raw data has been supplied as a Supplemental Dataset.

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
