# Peer review of "The inheritance of female colour polymorphism in Ischnura genei (Zygoptera: Coenagrionidae), with observations on melanism under laboratory conditions"

_PeerJ, doi:10.7717/peerj.2380_

## Round 0.1 · original submission · Major Revisions

Please note that in addition to the comments by the referees, I have also included some items that need clarification/editing. These are:

Line 19: Rewrite to: “…exhibited different levels of melanism…”
Line 53: Capitalize “island”
Line 54: Clarify what you mean by “good species”
Line 56: Clarify what you mean by “colour maturation”
Line 64: Clarify what you mean by infuscans and aurantiaca
Line 70: Clarify what you mean by K,S andT
Line 78: Spring water was used.
Line 80: Needs more clarification: How long was the stick? Diamter was? Was it from the wild? How was it placed in the cage?
Line 81: Why was water removed? How much?
Lines 82-84: Needs clarification.
Line 85: Replace “…were employed as…” with “provided”
Line 88: Please explain the use of D. melanogaster. this comes out of nowhere.
Line 113: Change “was” to “were”
Line 129: Please re-write/clarify this sentence: “Between five and seven males did not respond to each model.”
Line 133: Please explain what terms were entered into the model.
Line 134: Why not analyze gertility data using GLM with appropriate errors?
Line 141: Please explain why you calculated a dispersion parameter. What does it tell you?
Line 142: Change “…including model phenotype and model identity (each specimen used in the experiment) as predictor variables.” to “…with model phenotype and model identity (each specimen used in the experiment) enrtered as predictor variables.”
Lines 157-158: Shouldn’t this information be given in the Introduction? Seems odd to use terminology throught a manuscript only to define it at the end.
Line 185: Change “apparition” to “appearance”
Line 264: Change to “determine”

·

Basic reporting

First paragraph of the introduction needs to be improved. There are other mechanisms maintaining phenotypic variation, including phenotypic plasticity and mechanisms maintaining genetic polymorphisms other than frequency- and density-dependence l. (28-30). The authors should take more effort to demonstrate how the work fits into the broader field of knowledge – as for now this part is rather superficial.

Experimental design

Justification for the study given in line 49 is unconvincing: comparative analysis of colour polymorphism is better suited to resolve whether it is polyphyletic than determining dominance of alleles in different species.

Some parts of methods' description are unclear:

l. 137 Clarify retroactive testing – what fro and how was it done for?

l. 142, 226-233 Clarify the analysis of morph behavior: you recorded several behaviors, but you report a single result of logistic regression analysis. Whatever was done, it appears to be a loss of information, as from description is seems the behaviors could be ordered from the least to the most responsive.

Tables 2 3 and relavant parts of methods and results: genotypes were unknown, so the authors need to describe how they inferred them; I assume on the basis of phenotypes, but table 2 does not contain male phenotypes;

Validity of the findings

The authors should not only show that genotype frequencies are consistent with the expected model of inheritance, but also that they are no consistent with alternative models

Furthermore, its hard to judge the validity of the model presentation part, as it is not cleear how behavioral data were analysed (see above)

Additional comments

Other comments:
Fisher and Mother plots (see eg. Radwan 1995 Heredity, Fig. 1) are synthetic and much more effective way of presenting the data from crosses than a big tables 2 and 3.

l. 17 reads as if N=42 referred to one family with unexpected offspring ratios, needs to be modified

l. 354 The sentence is unclear.

Reviewer 2 ·

Basic reporting

Discussion should be shortened and you should avoid to repeat results in the discussion section.

Experimental design

In general the experimental design is well designed. My only concern is in the the line 64. Do this small simple size may affect your results? For example this may explain your results exposed in lines 256-258 or another result as line 300 regarding the unknown genetic mechanism.

Validity of the findings

No comments

Additional comments

Line 54. A good species? Please provide a better description or define this word "good species".

279-280. Could this happen in nature and animals may die before adulthood? Melanism impose a cost in autoimmunity an oxidative stress damage.

Line 353 …and what do you expect?

Could you expand a little bit more the impact of your result on the general literature of sexual polymorphism?

---

## Round 0.2 · Minor Revisions

Thank you for submitting a revised version of your manuscript. Although the current version is a superior manuscript to the first, I have found that it still needs a bit of editing to improve clarity and grammar. I have made my editorial changes as "track changes" in your document. I would like you to check if my edits are approproite as well as address my comments. My .docx document will be sent you under separate cover.

---

## Round 0.3 · accepted · Accept

Thank you very much for taking care of the my editorial suggestions.

The manuscript is now Accepted, but I noticed a small number of typos which you should fix in Production:

(Line numbers from your most recently submitted word doc):

Line 441/442: Replace the "undistinguishable" wording with "mature aurantiaca and infuscans females cannot be distinguished by males."

Line 199: Eighth is misspelt as eigth in the sentence "black spot on the eigth abdominal segment."

Line 418 (the sentence: "or if their follow different strategies to lay a different number of eggs through time ") - the word "their" should be "they"